# Uncovering the mechanism linking education expectation and suicide ideation among students in tertiary education: The mediating role of academic burnout, psychache, and hopelessness

Mohammad Izzat Morshidi[1]*, Peter K. H. Chew[2⊙], Lidia Suarez[2⊙]

1 Department of Psychology, University of Portsmouth, United Kingdom, 2 School of Social and Health Sciences, James Cook University Singapore, Singapore

⊙ These authors contributed equally to this work.
* Izzat.morshidi@port.ac.uk

## Abstract

High educational expectations are known risk factors for poor psychological well-being and suicide risk among tertiary students. However, the underlying mechanism linking high educational expectations to suicide risk is unclear. Additionally, it is also uncertain if suicide risk due to high expectations is universal or culture specific. The current study proposes a model examining the mediating role of academic burnout, psychache, and hopelessness in explaining the link between educational expectations and suicide ideation. The study also examined the model across samples from Malaysia, Singapore, and Australia to examine cultural variances. The study involved 641 tertiary students from Malaysia, Singapore, and Australia who completed self-report measures on perceived educational expectation, psychache, hopelessness, academic burnout, and suicidal ideation. The model was analysed using path analysis on AMOS 28. Results revealed that academic burnout was not a significant simple mediator, and the mediating roles of psychache and hopelessness across sources of expectations and samples were mixed. Results revealed a significant mediating effect of academic burnout through psychache and hopelessness on suicide ideation for models with self-expectation and parental expectation in all samples. Overall, our study highlights the complex mechanisms underlying the connection between different sources of educational expectations onto suicide ideation among students in higher education.

## Introduction

Suicide, defined as an act carried out by an individual with the expectation of a fatal outcome (death), ranks among the top five causes of mortality among youths globally [1]. Youth suicide is more troubling when we consider the rates of student suicide.

**Data availability statement:** The datasets and output supporting the conclusions of this article are available in figshare: https://figshare.com/projects/Education_expectation_and_suicide_ideation/217090.

**Funding:** The author(s) received no specific funding for this work.

**Competing interests:** The authors have declared that no competing interests exist.

An analysis of student samples from 36 tertiary education institutions (16 from North America, 12 from Asia, 6 from Europe, and 2 from Africa) found a lifetime prevalence rate of 22.3% for suicidal ideation and 3.2% for suicide attempts and a 12-month prevalence rate of 10.62% for suicidal ideation and 1.2% for suicide attempts [2]. Similarly, a review of 46 articles from 40 countries reported a prevalence of 18.8% for suicide ideation among university students [3]. A cross-national study of colleges in 11 Muslim countries reported that 34% of students reported having suicidal ideation during their studies [4]. These studies highlight a concerning rate of student suicides globally.

Suicide is also a leading cause of mortality among youths in the Western Pacific. The World Health Organization revealed a suicide rate of 17.18 per 100,000 among 15–29-year-olds in the Western Pacific based on data collected in 2021 [5]. Malaysia, Singapore, and Australia are examples of multicultural societies in the Western Pacific with concerning rates of youth suicide. In Malaysia, there is an increased rate of suicide attempts, planning, and thoughts among youth (13–17 years old) between the years 2011 and 2012 [6]. In Singapore, suicide was the leading cause of death among those aged 10–29 years old in 2019 [7]. The Australian Institute of Health and Welfare [8] recorded a total of 384 suicide deaths among 18–24-year-olds in 2019.

Several studies have reported that the risk of suicidal thoughts and behaviour among students varies across cultures [9,10]. A review of suicide deaths among youths (15–19 years) across 90 countries found that suicides in European nations were significantly lower than in non-European nations [11]. In another comparison, Snowdon et al. [12] found that the rate of suicides among youths between the years 2009 and 2013 was higher in Japan than in the United States. The study also found that suicide rates among male students were higher in Australia than in Hong Kong, while suicide rates among female students were higher in Hong Kong than in Australia [12]. The disparity in youth and student suicide rates between the East and West may point to the differences in the pressure to meet educational expectations.

### Educational expectations

Educational expectations refer to the belief towards the accomplishment of academic short-term and long-term goals. Morshidi et al. [13] identified four main sources of educational expectations perceived by students in tertiary education: the self, parents, educators, and culture. An examination of suicide notes in Singapore between the years 2000 and 2004 found that most suicide notes left by younger victims often cited school-related reasons such as not meeting high educational expectations and exam pressures [14]. Students who feel that they are incapable of fulfilling expectations reported an increased risk of psychological distress, shame, embarrassment, hopelessness [15], burnout [16], and suicide [17,18]. When examined further, the sources of educational expectation can be classified into two distinct categories: internal expectations and external expectations.

Internal expectations are developed and imposed by oneself, while external expectations are developed and imposed by others, such as family, friends, educators, the academic institution, or culture. Several studies have noted a significant link

between different sources of educational expectations on student mental health. High self-expectations have been associated with better motivation and academic achievement among students [19,20] but may also lead to increased distress [21]. Expectations from parents and family members have been linked to increased psychological distress [22–24] and suicidal tendencies [25]. For example, students who enrol in university due to parental pressure report greater stress than those who enrol willingly [26]. Educators also impose high expectations that negatively impact a student's self-esteem [19] and is positively associated with academic distress and burnout [27,28]. The sociocultural environment is a macro source of expectations. For example, the Confucian Heritage Culture that permeates most of East Asia places great importance on academic success as a means of upward social mobility and a child's obligation toward their family [29,30]. This theory has been used to explain why Asian or ethic Asian students experience a greater degree of academic pressure and a higher risk of poor mental health than their Western counterparts [31]. While the link between educational expectations, poor mental health, and suicide risk is well established, the underlying mechanism between expectations and suicide remains unclear. Not all students who are under intense educational expectations develop suicidal tendencies. It is also unclear which source of expectations presents a greater risk and if the risk varies across cultures.

## Academic burnout

Academic burnout is psychological distress characterized by emotional exhaustion from academic demands, feeling cynical about studying, and a perceived inadequacy as a student [16,32]. Increased academic burnout is associated with poor academic performance and cognitive functioning [33]. A meta-analysis found that the adverse effect of academic burnout on performance is robust across different educational levels [34]. Several studies have also found a positive association between academic burnout and suicidal ideation [35,36].

Burnout is a result of a discrepancy between a task and the capacity to complete the task [37]. The Job Demand-Control (JDC) model postulates that an individual is likely to experience distress when there is high demand but low autonomy [38]. The increased pressure to fulfil high expectations and demands imposed by oneself and others requires the exertion of prolonged effort, which can be emotionally, physically, and psychologically taxing to students [39]. Moreover, expectations that are imposed by others reflect a poor sense of autonomy, which is likely to cause exhaustion and cynicism [40]. In contrast, expectations from the self reflect greater autonomy and therefore a reduced risk of distress and exhaustion [41]. As such, we propose that academic burnout may explain the link between educational expectations and suicidal ideation.

## The Three-step theory of suicide

The Three-step Theory (3ST) of suicide [42] views suicide as a multi-dimensional phenomenon. Step one suggests that passive suicidal ideation is formed from an intense experience of psychological pain or psychache and hopelessness. The experience of psychache can manifest as feelings of depression, guilt, shame, defeat, or fatigue, which decreases the desire to continue living in a painful existence. Hopelessness is necessary as someone who perceives that the psychache may go away or improve will not develop thoughts of escape by suicide compared to someone who perceives they will gain no respite. In step two, suicidal ideation intensifies when someone has a poor sense of connectedness that provides meaning to one's life, thus leading the person to feel like their life is not worth living [43,44]. In step three, suicide ideation transitions into an attempt when the capacity for suicide is present. The capacity for suicide includes dispositional capacity (e.g., high pain tolerance), acquired capacity (e.g., history of abuse), and practical capacity (e.g., access to lethal means) [42]. The presence of a suicide capacity increases the likelihood of an attempt and the risk of death.

## The present study

Youth suicide is preventable, and insight into the mechanisms of suicide development is necessary to inform research and prevention. High expectations can lead to negative psychological outcomes [19,25,26], which are manifestations of

psychological pain and a sense of hopelessness [45]. We also identified academic burnout as both an outcome of high educational expectations (i.e., a demand beyond a student's control) and a predictor of suicide ideation per the JDC [38]. As academic burnout is characterized by feelings of emotional exhaustion, cynicism, and inadequacy, we propose that academic burnout also predicts psychache and hopelessness, subsequently predicting suicidal ideation. Cynical beliefs about the purpose of studying along with the perceived incompetence as a student, signify a pessimistic outlook and a sense of hopelessness toward their academic pursuit [16]. As such, the present study proposes a novel model that integrates the 3ST and the JDC in an attempt to examine the mechanism between educational expectations and suicide ideation through a hypothesised serial mediation model. Through this model, we empirically identified how burnout, psychological pain, and hopelessness predict suicide risk with the goal of informing targeted suicide prevention in mitigating the burden of educational expectations among students in tertiary education.

We predicted that the relationship between educational expectations and suicide ideation is mediated by i) psychache, ii) hopelessness, and iii) academic burnout respectively. We also predicted a serial mediating effect of iv) academic burnout and psychache and v) academic burnout and hopelessness on the relationship between educational expectations and suicide ideation. The model was tested across four sources of expectations (i.e., self, parents, educator/institution, culture), and three sample groups (Malaysia, Singapore, Australia) to examine cross-cultural variations (Fig 1).

## Method

### Participants

Students who were 18 years and older and were actively enrolled in Malaysian, Singaporean, and Australian universities were sampled through convenience and snowball sampling between July 2022 and December 2022. For the Malaysian sample, 380 responses were collected, where 156 responses were omitted due to missing data, resulting in a final 224 responses (77.7% female, 21% male, and 1.3% prefer not to say). For the Singaporean sample, 261 responses were collected, where 61 responses were omitted due to missing data, resulting in a final 200 responses (77.5% female, 21.5% male, and 1.0% prefer not to say). For the Australian group, 428 responses were collected, where 211 responses were omitted due to missing data, resulting in a final 217 responses (74.2% female, 23.5% male, and 2.3% prefer not to say). The final sample size for each country was adequate for a moderate effect size in regression models [46]. Further demographic details can be found in Table 1.

### Instruments

**Perceived educational expectations.** The Higher Educational Expectation Scale [13] is a 28-item scale that examines perceived educational expectations among tertiary students across four domains: self, parents, educators, and culture using a 5-point Likert scale ranging from 1 (*Strongly agree*) to 5 (*Strongly disagree*), with higher scores indicating

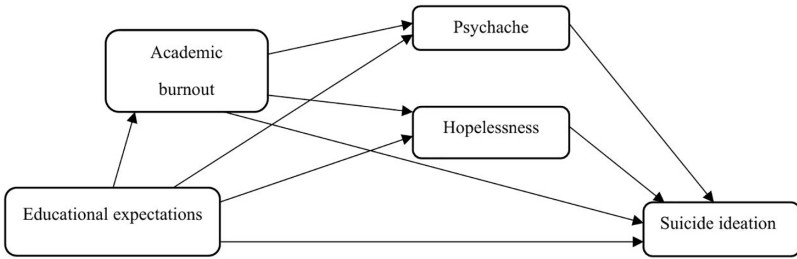

**Fig 1. Hypothesized serial mediation model.**

**Table 1. Participant demographic details.**

| Characteristics | Malaysia (*n* = 224) | Singapore (*n* = 200) | Australia (*n* = 217) |
|---|---|---|---|
| Age (*M, SD*) | 21.14 ± 2.59 | 21.96 ± 5.59 | 21.12 ± 3.04 |
| **Type of Higher Education Institution** | | | |
| Public | 18.30% | 32% | 72.80% |
| Private | 81.70% | 68% | 27.30% |
| **Course of Study** | | | |
| Foundation/Pre-University | 28.10% | 1% | 0.50% |
| Diploma/Polytechnic | 1.30% | 7.50% | 5.50% |
| Undergraduate/Bachelor's degree | 66.50% | 90% | 89.90% |
| Others (e.g., Master's, Doctorate) | 4.00% | 1.50% | 4.10% |

higher perceived expectations. The scale has good internal consistency, Cronbach's $\alpha$ = .70 [13] and shows excellent reliability in this study among samples from Malaysia ($\alpha$ = .91), Singapore ($\alpha$ = .91), and Australia ($\alpha$ = .91).

**Psychache.** The Scale of Psychache [47] is a 13-item scale that examines the degree of psychological pain. Items are rated on a 5-point Likert scale ranging from 1 (*Never or strongly disagree*) to 5 (*Always or strongly agree*), with higher scores indicating greater psychological pain. The scale has good internal consistency, Cronbach's $\alpha$ = .92 [47] and shows excellent reliability in this study among samples from Malaysia ($\alpha$ = .96), Singapore ($\alpha$ = .95), and Australia ($\alpha$ = .96)

**Hopelessness.** The Brief Hopelessness Scale [48] is a 6-item scale that measures hopelessness. Items are rated by either 'agree' (scored 1) or 'disagree' (scored 0). Responses are summed with higher scores indicating a higher degree of hopelessness, with good internal consistency Cronbach's $\alpha$ = .75 [48] and show excellent reliability in this study among samples from Malaysia ($\alpha$ = .84), Singapore ($\alpha$ = .84), and Australia ($\alpha$ = .83).

**Academic burnout.** Study Burnout Inventory [49] is a 9-item scale that examines burnout in higher education across three dimensions: exhaustion, cynicism, and inadequacy. Items are rated on a 6-point Likert scale ranging from 1 (*Completely disagree*) to 6 (*Completely agree*), with higher scores indicating higher degrees of study burnout with good internal consistency with ranges from Cronbach's $\alpha$ = .83 to.85 [49] and show excellent reliability in this study among samples from Malaysia ($\alpha$ = .89), Singapore ($\alpha$ = .87), and Australia ($\alpha$ = .89)

**Suicide ideation.** The Suicide Ideation Scale [50] is a 10-item scale that examines suicidal ideation across two domains: passive and active using a 5-point Likert scale ranging from 1 (*Never or none of the time*) to 5 (*Always or a great many times*), with higher cumulative scores indicating more intense suicidal ideation. The scale has good internal consistency, Cronbach's $\alpha$ = .86 [51] and shows excellent reliability in this study among samples from Malaysia ($\alpha$ = .92), Singapore ($\alpha$ = .94), and Australia ($\alpha$ = .95)

## Procedure

This cross-sectional study was hosted on Qualtrics, where participants were presented with the study information page, consent form, the five self-report instruments, and a participant support document that included details for free psychological support services in Malaysia, Singapore, and Australia, respectively. The study was advertised and disseminated through online student forums, social media, and email correspondences to higher education institutions in Malaysia, Singapore, and Australia. Given the sensitive nature of the study, participants were made aware of the risks of distress in the Information document and were provided with contact details of free psychological support services in respective countries in the support document. The 15-minute study was in English and approved by the James Cook University Human Research Ethics Committee (H8552).

## Proposed analysis

The study employed a structural equation modelling with the maximum likelihood estimation using the Statistical Package for Social Science 27.0 AMOS 28. We specified the model with one exogenous variable (educational expectations), four endogenous variables (academic burnout, psychache, hopelessness, suicide ideation), and nine pathways based on the postulations by the 3ST and JDC theory, resulting in an over-identified model (Fig 2). The same form (model) was tested with data for each source of educational expectation (self, parents, educator/institution, culture), respectively while data for the other variables remained constant. Testing the model for the different sources of expectations allows us to better identify how each source influences poor mental health outcomes. Multigroup analysis was performed for each model with Group 1 (Malaysia), Group 2 (Singapore), and Group 3 (Australia). Estimates of global model fit for each model include the comparative fit index (CFI), normed fit index (NFI), and goodness-of-fit statistic (GFI) values near or greater than 0.95, and a root mean square error of approximation (RMSEA) value near or less than 0.06 [52].

A pairwise parameter comparison using the critical ratio for differences on the nine specified pathways was examined. For a two-tailed test, a critical ratio difference z-score between the critical range of ±1.96 indicates no significant difference. Multigroup analysis allows us to determine cross-cultural variances in the effect of different types of expectations on poor mental health outcomes.

## Results

### Preliminary analysis

Prior to analysis, the dataset was screened. Incomplete responses were treated as missing with no imputation method used and were dropped from the final dataset. Note that possible bias may occur due to deletion by not accounting for the reasons for missingness. A one-way analysis of variance was conducted to examine the differences between the four sources of expectation scores and variables across the Malaysian, Singaporean, and Australian samples (Table 2). There was a significant difference between the samples for each source of expectation. Post-hoc analysis using Bonferroni correction revealed that for expectations from self, the mean for Malaysia was significantly higher than Singapore ($t = 2.979$, $p = .009$). For expectations from parents, the mean for Malaysia was significantly higher than Singapore ($t = 2.484$, $p = .040$) and Australia ($t = 4.724$, $p < .001$). For expectations from educators, the mean for Malaysian was significantly higher than Singapore ($t = 2.904$, $p = .011$) and Australia ($t = 4.263$, $p < .001$). For expectations from culture, the mean for Malaysia was significantly higher than Singapore ($t = 2.929$, $p = .011$) and Australia ($t = 8.398$, $p < .001$). Moreover, the mean expectation from culture for Singapore was significantly higher than Australia ($t = 5.254$, $p < .001$). There were no significant differences between the samples on measures of academic burnout, psychological pain, and suicide ideation. Only hopelessness was significantly different across samples, $F(2, 638) = 3.641$, $p = .027$. Post-hoc analysis using Bonferroni correction revealed that mean hopelessness was significantly higher for Malaysia than Australia ($t = 2.569$, $p = .031$).

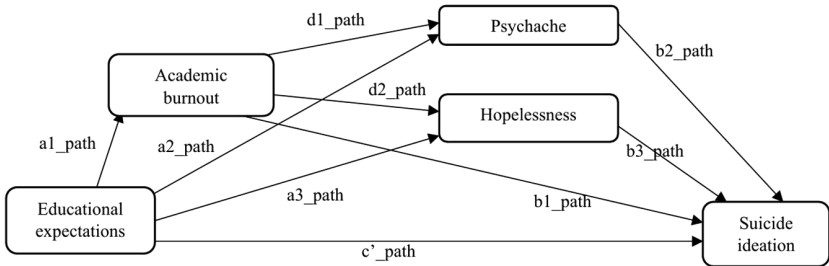

**Fig 2. Structural serial mediation model with defined pathways.**

**Table 2. Differences across samples.**

| Source of expectation | Sample group | M | SD | F |
|---|---|---|---|---|
| Self | Malaysia | **46.04** | 8.498 | 4.460* |
| | Singapore | 43.54 | 9.188 | |
| | Australia | 44.71 | 8.262 | |
| Parents | Malaysia | **15.65** | 4.315 | 11.188** |
| | Singapore | 14.52 | 4.642 | |
| | Australia | 13.54 | 5.075 | |
| Educators | Malaysia | **17.21** | 3.658 | 9.556** |
| | Singapore | 16.18 | 3.534 | |
| | Australia | 15.73 | 3.776 | |
| Culture | Malaysia | **22.97** | 4.525 | 36.078** |
| | Singapore | **21.46** | 5.144 | |
| | Australia | **18.73** | 6.108 | |
| Academic Burnout | Malaysia | 32.27 | 9.588 | .189 |
| | Singapore | 31.95 | 9.249 | |
| | Australia | .31.71 | 9.838 | |
| Psychache | Malaysia | 20.03 | 12.160 | .144 |
| | Singapore | 19.83 | 11.413 | |
| | Australia | 19.46 | 11.665 | |
| Hopelessness | Malaysia | **1.46** | 1.908 | 3.641* |
| | Singapore | 1.37 | 1.857 | |
| | Australia | **1.01** | 1.637 | |
| Suicide ideation | Malaysia | 16.31 | 7.474 | 1.270 |
| | Singapore | 16.84 | 7.648 | |
| | Australia | 15.67 | 7.464 | |

*Note*. Bolded values indicate mean scores that are different. *$p$ < .05, **$p$ < .001.

## Serial mediation analysis

**The self expectation model.** We tested the specified model with the exogenous variable self expectation (Fig 3). Results found no significant indirect mediating effect of burnout for Group 1 ($\beta$ = .002, 95% percentile CI [−.014,.019]), Group 2 ($\beta$ = .008, 95% percentile CI [−.005,.032]), and Group 3 ($\beta$ = .007, 95% percentile CI [−.007,.028]). There was also no significant indirect mediating effect of psychache for Group 1 ($\beta$ = .029, 95% percentile CI [−.017,.088]), Group 2 ($\beta$ = .050, 95% percentile CI [−.004,.127]), and Group 3 ($\beta$ = .024, 95% percentile CI [−.026,.082]). No significant indirect mediating effect was also found for hopelessness in Group 1 ($\beta$ = −.026, 95% percentile CI [−.063,.005]), Group2 ($\beta$ = −.017, 95% percentile CI [−.047,.000]), and Group 3 ($\beta$ = −.019, 95% percentile CI [−.055,.007]).

A significant negative indirect serial mediating effect of academic burnout through psychache was found in Group 1 ($\beta$ = −.042, 95% percentile CI [−.080, −.017]), Group 2 ($\beta$ = −.053, 95% percentile CI [−.105, −.019]), and Group 3 ($\beta$ = −.053, 95% percentile CI [−.098, −.019]). Similarly, a significant negative indirect serial mediating effect of academic burnout through hopelessness was found in Group 1 ($\beta$ = −.017, 95% percentile CI [−.038, −.006]), Group 2 ($\beta$ = −.010, 95% percentile CI [−.029, −.003]), and Group 3 ($\beta$ = −.016, 95% percentile CI [−.039, −.005]). Global model fit was mixed, CFI = .884, NFI = .884, GFI = .942, and RMSEA = .233.

**The parental expectation model.** We tested the specified model with the exogenous variable parental expectation (Fig 4). Results found no significant mediating effect of burnout in Group 1 ($\beta$ = .000, 95% percentile

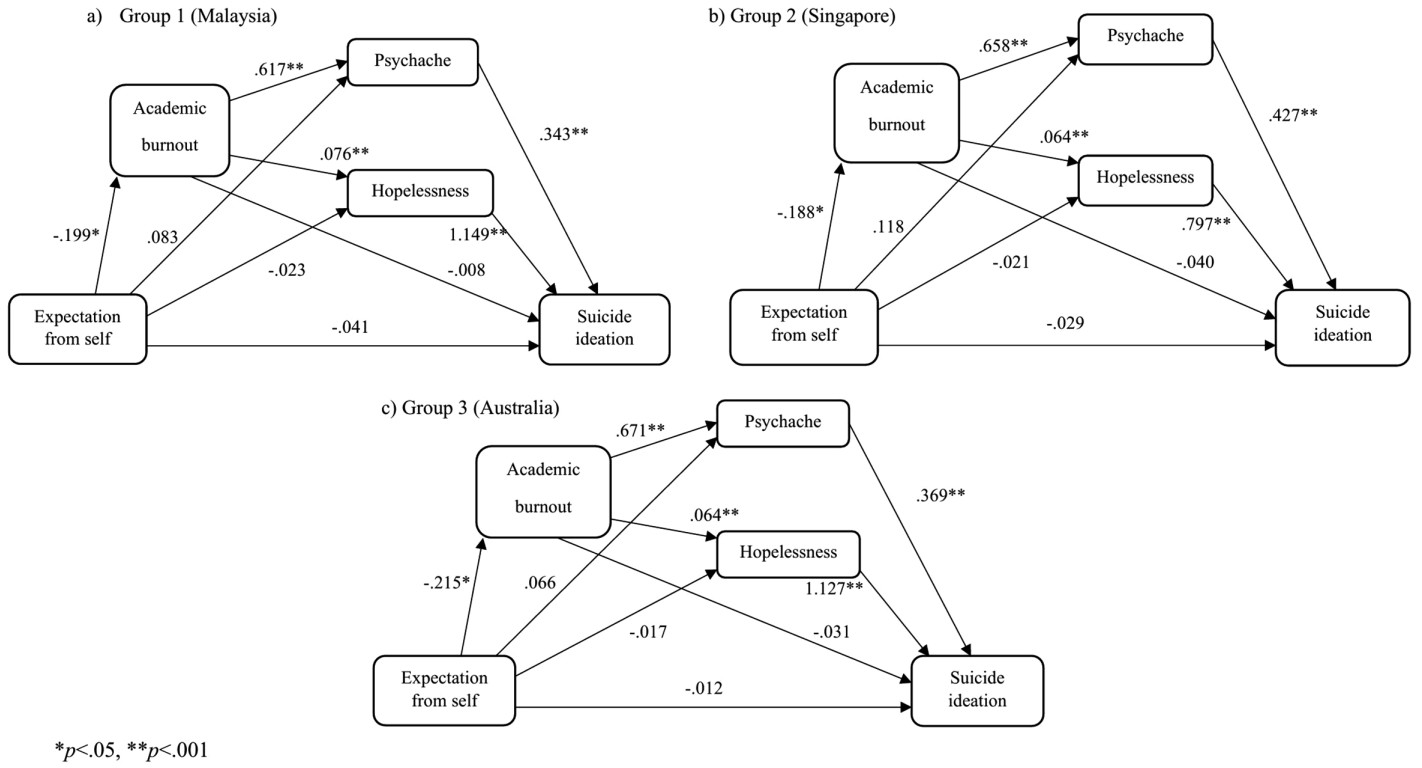

*p<.05, **p<.001

**Fig 3. Self expectation models.**

CI [−.039,.040]), Group 2 ($\beta = −.018$, 95% percentile CI [−.066,.022]), and Group 3 ($\beta = −.010$, 95% percentile CI [−.055,.030]). There was a significant full mediating effect of psychache for Group 1 ($\beta = .260$, 95% percentile CI [.162,.379]) but not for Group 2 ($\beta = .083$, 95% percentile CI [−.020,.212]) and Group 3 ($\beta = .099$, 95% percentile CI [.002,.207]). There was also a significant full mediating effect of hopelessness for Group1 ($\beta = .095$, 95% percentile CI [.038,.169]) but not for Group 2 ($\beta = −.010$, 95% percentile CI [−.038,.056]), and Group 3 ($\beta = .042$, 95% percentile CI [−.001,.104]).

However, a significant indirect serial mediating effect of academic burnout through psychache was found in Group 1 ($\beta = .088$, 95% percentile CI [.030,.154]), Group 2 ($\beta = .144$, 95% percentile CI [.079,.235]), and Group 3 ($\beta = .128$, 95% percentile CI [.081,.202]). Similarly, a significant indirect serial mediating effect of academic burnout through hopelessness was found in Group 1 ($\beta = .041$, 95% percentile CI [.016,.078]), Group 2 ($\beta = .031$, 95% percentile CI [.011,.074]), and Group 3 ($\beta = .040$, 95% percentile CI [.017,.083]). Global model fit was mixed, CFI = .904, NFI = .903, GFI = .949, and RMSEA = .217.

**The Educator/Institutional expectation model.** We tested the specified model with the exogenous variable educator/institution expectation (Fig 5). Results found no significant indirect mediating effect of burnout for Group 1 ($\beta = .000$, 95% percentile CI [−.026,.021]), Group 2 ($\beta = .000$, 95% percentile CI [−.026,.021]), and Group 3 ($\beta = −.009$, 95% percentile CI [−.054,.009]). There was a significant full mediating effect of psychache for Group 1($\beta = .157$, 95% percentile CI [.034,.273]) and Group 3($\beta = .159$, 95% percentile CI [.071,.274]) but not for Group 2 ($\beta = .117$, 95% percentile CI [−.020,.295]). There was no significant mediating effect of hopelessness for Group 1 ($\beta = .033$, 95% percentile CI [−.013,.085]), Group 2 ($\beta = .030$, 95% percentile CI [−.008,.099]), and Group 3 ($\beta = −.025$, 95% percentile CI [−.027,.092]).

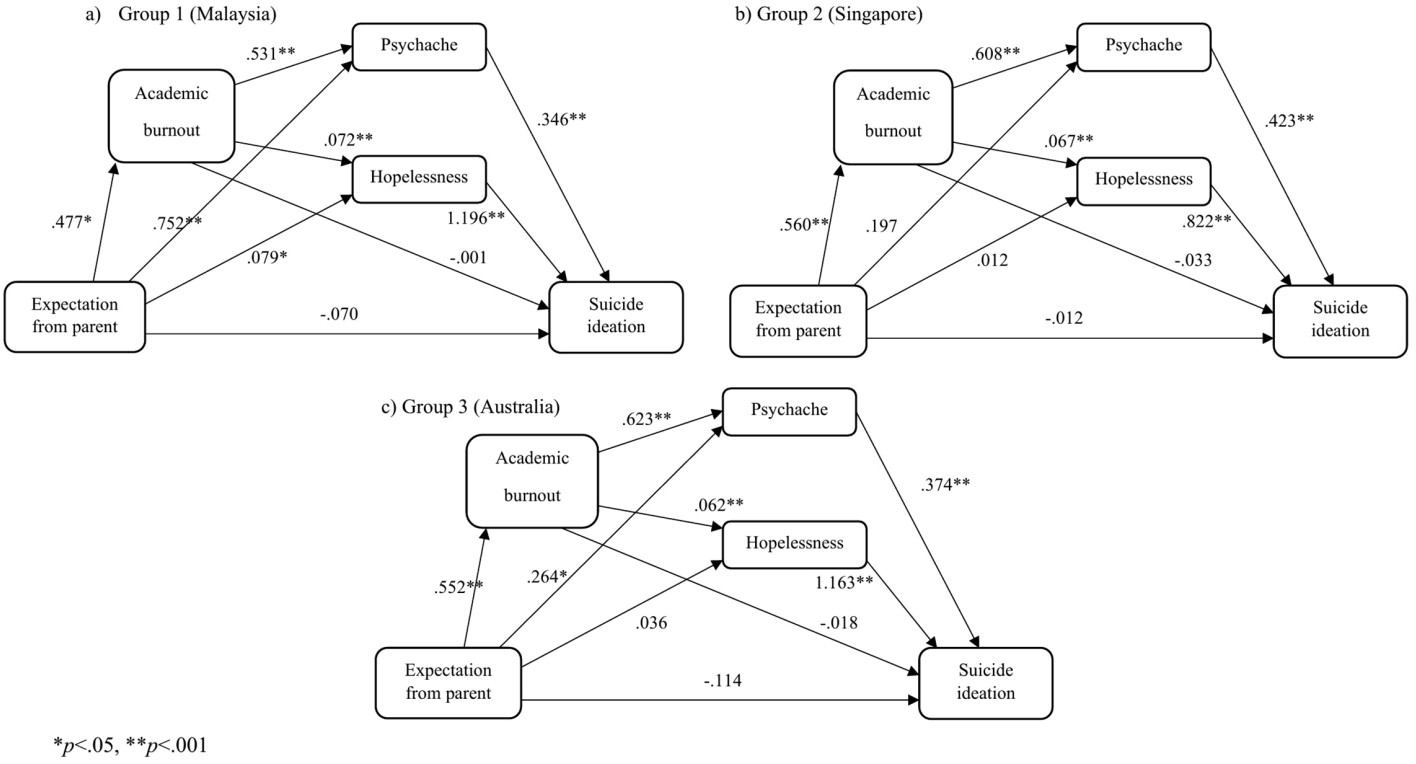

**Fig 4. Parental expectation models.**

There was also no significant serial mediating effect of academic burnout through psychache in Group 1 ($\beta$ = .046, 95% percentile CI [−.014,.119]), Group 2 ($\beta$ = .002, 95% percentile CI [−.095,.115]), and Group 3 ($\beta$ = .078, 95% percentile CI [.002,.173]). Similarly, there was no significant serial mediating effect of academic burnout through hopelessness in Group 1 ($\beta$ = .021, 95% percentile CI [−.003,.056]), Group 2 ($\beta$ = .000, 95% percentile CI [−.019,.027]), and Group 3 ($\beta$ = .024, 95% percentile CI [.002,.067]). Global model fit was mixed, CFI = .890, NFI = .891, GFI = .946, and RMSEA = .224.

**The culture expectation model.** We tested the specified model with the exogenous variable cultural expectation (Fig 6). Results found no significant mediating effect of burnout for Group 1 ($\beta$ = .000, 95% percentile CI [−.017,.013]), Group 2 ($\beta$ = −.007, 95% percentile CI [−.038,.005]), and Group 3 ($\beta$ = −.008, 95% percentile CI [−.035,.010]). There was a significant full mediating effect of psychache in Group 1 ($\beta$ = .106, 95% percentile CI [.014,.200]), Group 2 ($\beta$ = .148, 95% percentile CI [.073,.239]), and Group 3 ($\beta$ = .134, 95% percentile CI [.064,.213]). However, a significant mediating effect of hopelessness in Group 3 ($\beta$ = .049, 95% percentile CI [.015,.098]) but not for Group 1($\beta$ = .036, 95% percentile CI [−.025,.099]) and Group 2 ($\beta$ = .024, 95% percentile CI [−.003,.074]).

Additionally, results show a significant serial mediating effect of burnout through psychache for Group 3 ($\beta$ = .063, 95% percentile CI [.026,.123]) but not for Group1 ($\beta$ = .025, 95% percentile CI [−.025,.086]) and Group 2 ($\beta$ = .051, 95% percentile CI [−.008,.118]). Similarly, results show a significant serial mediating effect of burnout through hopelessness for Group 3 ($\beta$ = .020, 95% percentile CI [.007,.047]) but not for Group 1 ($\beta$ = .011, 95% percentile CI [−.013,.039]) and Group 2 ($\beta$ = .011, 95% percentile CI [−.001,.031]). Global model fit was mixed, CFI = .898, NFI = .898, GFI = .948, and RMSEA = .218.

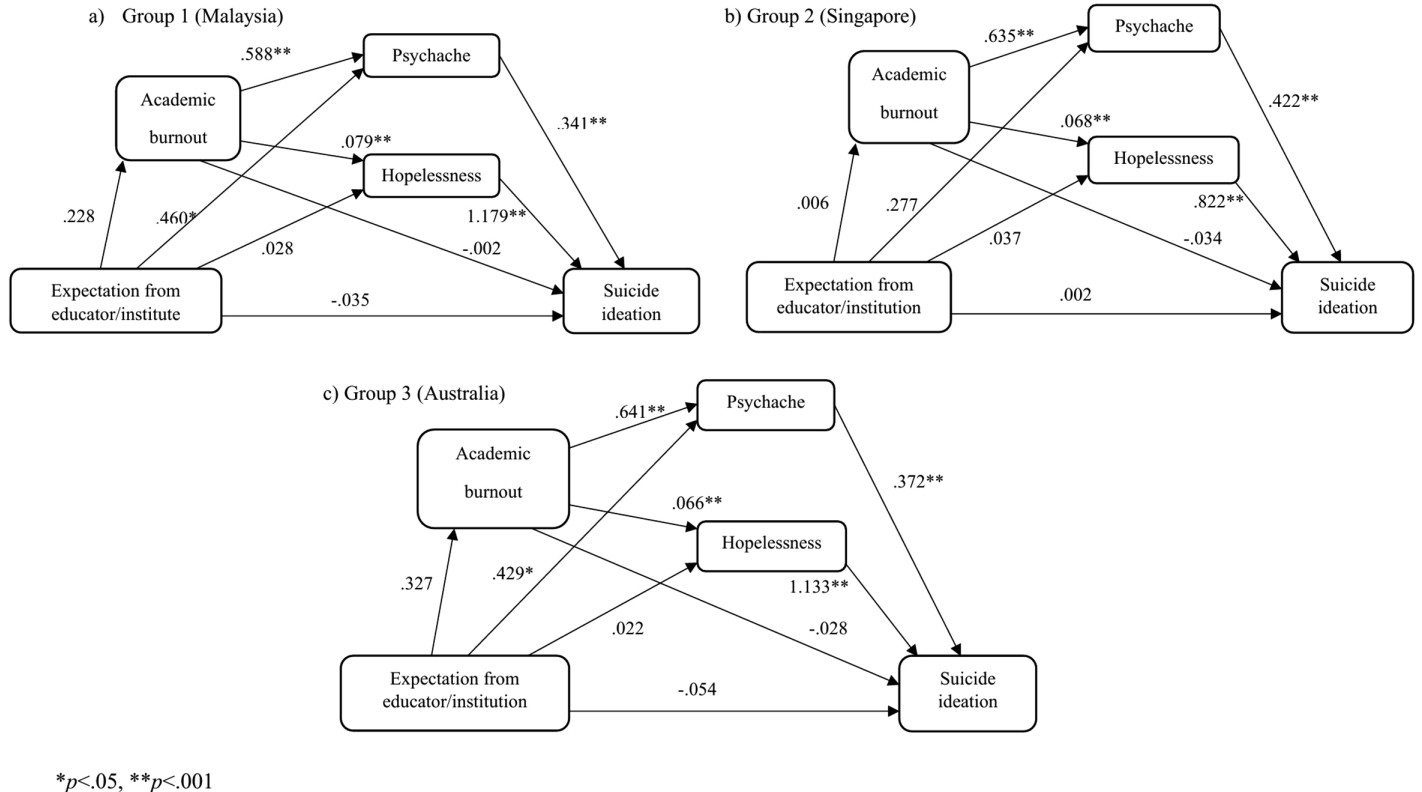

*p<.05, **p<.001

**Fig 5. Educator/Institutional expectation models.**

## Cultural variations in pathways

To examine cultural differences, an analysis using critical ratios for differences in the pathways comparing Group 1 (Malaysia), Group 2 (Singapore), and Group 3 (Australia) was conducted on AMOS 28 (see supplemental materials). For models with expectations from self, there were no significant differences in all pathways across groups. For models with expectations from parents, only the a2_path (Expectation from Parents à Psychache) was significantly different across groups (z-score above ±1.96). In particular, the differences were significant between the Malaysian and Singaporean models (z-score = −2.474) and between the Malaysian and Australian models (z-score = −2.318). No differences in the a2_path were found between the Singaporean and Australian models (z-score = .327). For models with expectations from educators, no significant differences were found in the pathways across groups. Similarly, for models with expectations from culture, no significant differences were found in the pathways across groups.

## Discussion

### Preliminary analysis

Preliminary findings revealed an overall significant difference in mean scores for the Malaysian sample on all four sources of expectations, which were higher than the Singapore and Australia samples. Differences between the Singaporean and Australian samples were not found for expectations from self, parents, and educators. Expectation from culture was significantly different between all groups, with Malaysia having the highest mean score, followed by Singapore and Australia with the lowest score. The result supports the argument that Eastern Confucian cultures (e.g., Malaysia and Singapore)

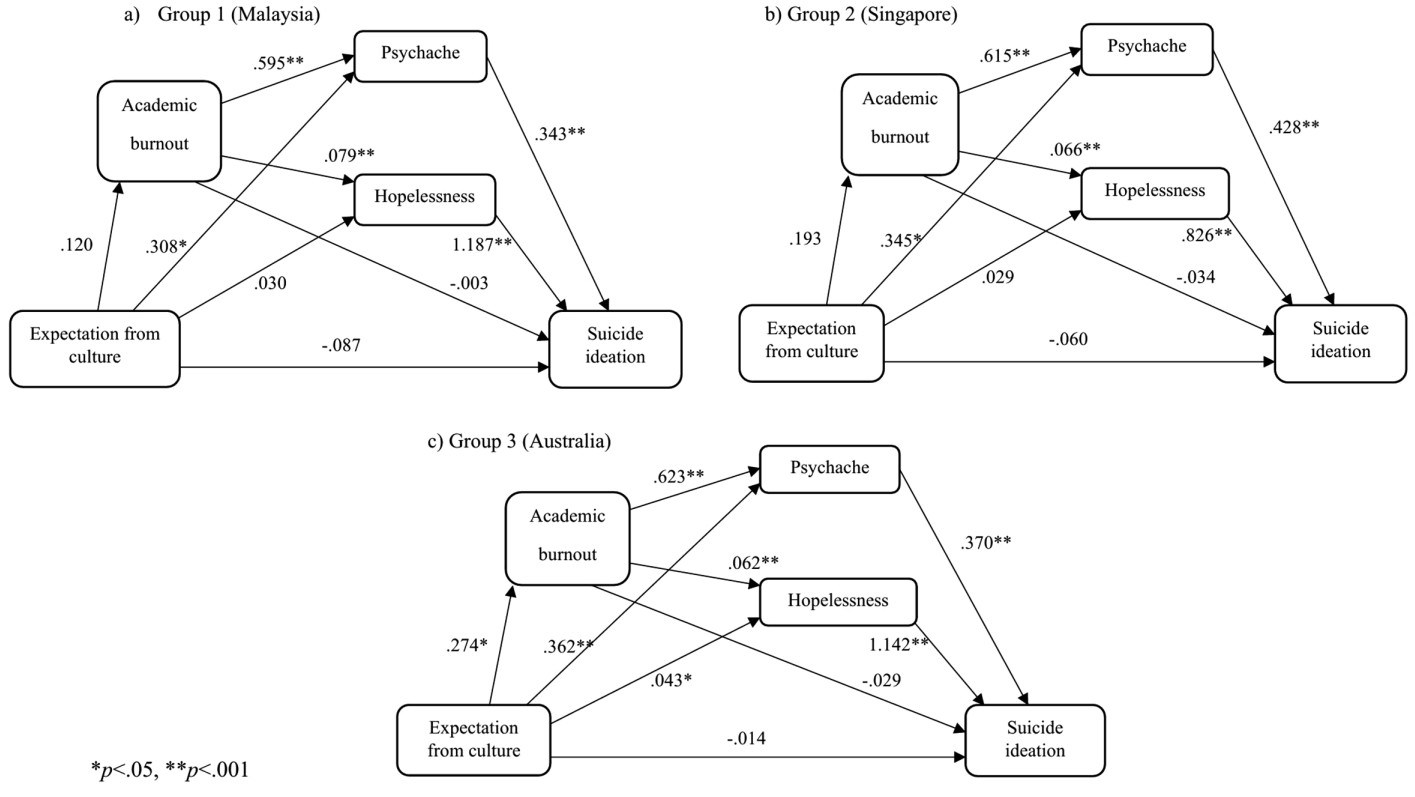

**Fig 6. Culture expectation models.**

place a greater emphasis on academic success and therefore impose greater educational expectations on students than Western cultures (e.g., Australia) [53,54].

## Analysis of the mediation models

None of the models provided evidence to support the mediating effect of academic burnout on suicide ideation across the different sources of expectation and samples. Upon further examination of the pathways in each sample, self expectation was consistently negatively linked to academic burnout. This negative link suggests that, unlike demands from external sources, self-developed expectations are within one's control and reflects one's actual capacity, which are less likely to be emotionally and psychologically taxing, thus reducing the risk of burnout [39,40]. Our results also did not find a link between academic burnout and suicidal ideation across all models, which contradicts findings of a positive association between academic burnout and suicide ideation [55,56]. We postulate that the link between academic burnout and suicide may not be direct. Rather, the link can be better explained through the mediating role of psychache and hopelessness. As such, academic burnout is not a simple mediator between educational expectations and suicide ideation.

For the second hypothesis, the mediating effect of psychache was mixed across different sources of expectation and samples. Psychache was a significant mediator for the models with the external expectation for the Malaysian sample only. However, psychache mediated the link between cultural expectations and suicide ideation across all samples, suggesting a robust effect of sociocultural educational pressure on poor psychological outcomes and suicide risk. Although mixed, our findings are consistent with research demonstrating a link between external sources of expectation and poor psychological outcomes [19,26,29]. Moreover, the significant associations between psychache and suicide ideation across

all models of expectations and samples support the salient effect of psychache in predicting suicidal ideation as postulated in the first step of the 3ST [42].

For the third hypothesis, a significant mediating effect of hopelessness was only found in models with the expectation from parents for the Malaysian sample and with the expectation from culture for the Australian sample. Overall, our results suggest that hopelessness does not mediate the relationship between educational expectations and suicide ideation. This is not surprising as studies have suggested that hopelessness is only present upon experiencing an unsatisfactory outcome [57,58]. As such, merely experiencing high expectations without prior experiences of failure is not sufficient to predict hopelessness. Additionally, the significant associations between hopelessness and suicide ideation across all models lend support to the first step of the 3ST [42].

For the fourth and fifth hypotheses, results found evidence for the serial mediating effect of academic burnout through psychache and academic burnout through hopelessness for models with expectations from self and parents across all samples. These findings affirm that academic burnout is not a simple mediator. Our findings suggest that suicide ideation is likely when a student experiences academic burnout that is accompanied by the feeling of intense psychological pain or a sense of hopelessness. The consistent serial mediating effect across samples also suggests that students universally perceive high expectations from parents as stressful. This challenges the narrative that students from an Asian background are more pressured by their parents than students from a Western background [31,59]. Instead, we argue that the negative impact of high parental expectations on a student's psychological well-being is not necessarily influenced by ethnicity or culture.

A negative effect was found in serial mediation models with self-expectations across samples. This suggests that expectations from self negatively predict academic burnout and subsequently reduce the risk of psychache, hopelessness, and suicide ideation. Perceived control over expectations may explain the differences in the serial mediating effect between self-expectation and parental expectation. Self-determination theory [60] suggests that an optimal state of psychological well-being, functioning, and intrinsic motivation is determined by the fulfilment of three basic psychological needs: connectedness, competence, and control. The sense of control involves having ownership and freedom over choices in life and being guided by personal values or interests [60]. The obstruction of a psychological need and personal goal results in psychache, which is a robust predictor of suicidal ideation [61]. Students with a greater sense of autonomy and self-determination are less likely to report hopelessness and suicidal ideation when faced with negative life events [62]. Self-expectations reflect a higher degree of autonomy as students can accurately gauge their academic capacity and adjust their goals accordingly, while parental expectations leave a student helpless as they are not able to influence or control these expectations, which can be unrealistic or beyond their ability. This is exemplified in Larcombe et al. [63] who found that students who pursued a law degree out of personal interest had lower depressive symptoms than students who enrolled for other reasons (i.e., parental advice, best option available). The study also noted that enrolling based on 'personal interest and aptitude' was protective against depressive symptoms. In sum, students with a greater sense of autonomy are adaptable and able to overcome feelings of being trapped (hopelessness) when faced with challenges in higher education than those with poorer perceived autonomy. The consistency of the negative effect of self-expectation on suicide ideation across samples demonstrates the universal need for autonomy as a precursor of positive psychological well-being in the face of educational expectations.

## Cross-cultural variations

When models for each source of expectations were compared across samples, only the pathway from parental expectation to psychache was significantly different. This suggests that psychache due to parental expectation is likely for Malaysian students but not for Singaporean and Australian students. The disparity may be explained by Malaysia's aspiration to be a developed nation, which may influence many aspects of a student's academic pursuit. Malaysian students may perceive that their parents place significantly greater academic pressure to be more accomplished than parents in Singapore

and Australia. Nonetheless, further research examining sociocultural variance and influence on parental expectations and its psychological impact should go beyond the Eastern-Western cultural dyad and explore other sociocultural elements (e.g., immigration status).

## Strengths and limitations

This study is the first to examine the mechanism linking educational expectations to suicide ideation among tertiary students based on theory, as most studies have only drawn associations between educational expectations and suicide. The study also utilized a multidimensional instrument to examine the effect of different sources of perceived expectations on suicide risk, in addition to exploring the effect of high expectations on suicide across three different cultures. Limitations include not accounting for the ethnic, racial, and cultural demography of the students in each sample group, as our study was only interested in the perceived expectations of tertiary students in each country. Given the sampling limitations, caution should be taken in generalizing the findings. Future research should utilize a more representative technique to sample university student populations. As suicide ideation was the outcome variable, there may also be inaccuracies in responses by participants, as suicide is considered taboo, especially in Malaysia and Singapore [64,65]. Furthermore, cross-cultural validation and reliability of the instruments used in this study across Malaysia, Singapore, and Australia need to be examined further.

## Conclusion

This study sought to explain the underlying mechanism between educational expectations and suicide ideation. Our findings revealed a significant serial mediating effect of academic burnout through psychache and hopelessness to suicide ideation for models with self-expectation and parental expectation across all samples. Therefore, we have demonstrated that there is no direct link between high educational expectations and suicide ideation. Rather, the link between educational expectations and suicide ideation is complex. Future studies should examine the interaction of factors such as personality and meaning in life toward the development of suicidal ideation. Similarly, alternative models that examine the moderating role of burnout, psychache, and hopelessness may help identify high-risk students and better inform suicide prevention efforts among tertiary students.

## Author contributions

**Conceptualization:** Izzat Morshidi.

**Data curation:** Izzat Morshidi.

**Formal analysis:** Izzat Morshidi.

**Investigation:** Izzat Morshidi.

**Methodology:** Izzat Morshidi.

**Software:** Peter K. H. Chew, Lidia Suarez.

**Supervision:** Peter K. H. Chew, Lidia Suarez.

**Writing – original draft:** Izzat Morshidi.

**Writing – review & editing:** Izzat Morshidi, Peter K. H. Chew, Lidia Suarez.

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
