## [Decision Letter · Decision Letter 0]

9 Jul 2024

Dear Dr. Morshidi,

Thank you for submitting your manuscript to PLOS ONE. After careful consideration, we feel that it has merit but does not fully meet PLOS ONE’s publication criteria as it currently stands. Therefore, we invite you to submit a revised version of the manuscript that addresses the points raised during the review process.

**ACADEMIC EDITOR: It is an interesting study that assessed the contemporary issues of suicide and related factors. However, we further require some correction on editorial issues as well as contents of the document. Up on revision please make your document more clear and easily understandable by readers.**

We look forward to receiving your revised manuscript.

Kind regards,

Yadeta Alemayehu

Academic Editor

PLOS ONE

Additional Editor Comments:

I have the following comments:-

1. Please make the objective of your study clear and easily understandable. It is not clear what you are going to do from your title?

2. Under introduction it is better to start with definition and burden of the problem world wide than of the specific country.

3. Your result and methodology seems to have no clear boundary. Some bio-demographic data are the under methodology section.

4. The major finding of the study should be included in the conclusion section than only recommending  researchers.

5. Some other components like abbreviations,... could be included in the document.

Reviewers' comments:

Reviewer's Responses to Questions

**Comments to the Author**

1. Is the manuscript technically sound, and do the data support the conclusions?

Reviewer #1: Yes

Reviewer #2: Yes

2. Has the statistical analysis been performed appropriately and rigorously?

Reviewer #1: I Don't Know

Reviewer #2: Yes

3. Have the authors made all data underlying the findings in their manuscript fully available?

Reviewer #1: Yes

Reviewer #2: Yes

4. Is the manuscript presented in an intelligible fashion and written in standard English?

Reviewer #1: Yes

Reviewer #2: Yes

Reviewer #1: This subject is of high importance due to the recent increased attempts and ideation of suicide by the students, globally.

I particularly liked the examining of the underlying mechanisms and relations between multi factors.

Reviewer #2: This is a very interesting study and has implications for both educators and counsellors who work with students, along with curriculum designers.

Although the authors have presented data regarding the sources of expectations on students, it was not clear if these actually led to suicidal ideation or if that was assumed or implied by the authors without supportive substantive data. Kindly clarify this aspect of your study.

**Do you want your identity to be public for this peer review?** For information about this choice, including consent withdrawal, please see our Privacy Policy

Reviewer #1: **Yes: ** Dr Lily Abedipour MD

Reviewer #2: No

---

## [Author Response · Author response to Decision Letter 1]

9 Aug 2024

RESPONSE TO EDITOR

1. Please make the objective of your study clear and easily understandable. It is not clear what you are going to do from your title?

We have revised the title to clearly indicate the objective of the study. The new title reads: Uncovering the mechanism linking educational expectations and suicide ideation among students in tertiary education: The mediating role of academic burnout, psychache, and hopelessness.

We have also revised the section on the Current Study (on line 140) to better clarify the aim of the study for ease of understanding.

2. Under introduction it is better to start with definition and burden of the problem world wide than of the specific country.

We believe we have sufficiently provided the burden of the problem regarding youth and student suicide on a global scale (i.e., Asian continent, Western Pacific) on line 48. To further support this, we have added details and a definition of suicide to the first paragraph as well.

3. Your result and methodology seems to have no clear boundary. Some bio-demographic data are the under methodology section.

Additional details regarding the inclusion criteria of the sample has been included (see line 166). This highlights the boundary regarding our participant selection. We opted to report demographic data in the Participant sub-section to allow the Result section to focus solely on the analysis of the hypothesized model.

4. The major finding of the study should be included in the conclusion section than only recommending researchers.

We have revised the conclusion and include a short summary of the major findings (see line 461).

5. Some other components like abbreviations,... could be included in the document.

We acknowledge the comment and feel that the current amount of abbreviations is sufficient to aid understanding of the manuscript’s content.

RESPONSE TO REVIEWERS

1. Is the manuscript technically sound, and do the data support the conclusions?

Reviewer #1: Yes

Reviewer #2: Yes

Thank you

2. Has the statistical analysis been performed appropriately and rigorously?

Reviewer #1: I Don't Know

Reviewer #2: Yes

Thank you

3. Have the authors made all data underlying the findings in their manuscript fully available?

Reviewer #1: Yes

Reviewer #2: Yes

Thank you

4. Is the manuscript presented in an intelligible fashion and written in standard English?

Reviewer #1: Yes

Reviewer #2: Yes

Thank you

5. Review Comments to the Author

Reviewer #1: This subject is of high importance due to the recent increased attempts and ideation of suicide by the students, globally. I particularly liked the examining of the underlying mechanisms and relations between multi factors.

Thank you

Reviewer #2: This is a very interesting study and has implications for both educators and counsellors who work with students, along with curriculum designers. Although the authors have presented data regarding the sources of expectations on students, it was not clear if these actually led to suicidal ideation or if that was assumed or implied by the authors without supportive substantive data. Kindly clarify this aspect of your study.

The study attempts to provide more insight into the narrative that high educational expectations can lead to suicidal outcomes among student (see line 80 to 85). In this study, we argue that high educational expectations do not necessarily lead to suicide ideation and that the link is more nuanced. We proposed that academic burnout, psychache, and hopelessness are mediating factors that can explain variability in the risk of suicide ideation due to high educational expectation (see line 140 onwards).

Data about suicidal ideation among students were collected using the Suicide Ideation Scale (see line 199). This scale provides data on the degree of suicidal ideation with higher scores indicating greater suicidal ideation than lower scores. Our analysis through mediation modelling, tests the predictability of educational expectation on suicide ideation by examining the influence of the mediators (i.e., academic burnout, psychache, hopelessness). In all our findings, the data suggests that educational expectations do not directly predict suicide ideation and that the link between educational expectations and suicide ideation varies according to the source of expectation and through mediating factors (see Results section).

6. PLOS authors have the option to publish the peer review history of their article (what does this mean?). If published, this will include your full peer review and any attached files. Do you want your identity to be public for this peer review? For information about this choice, including consent withdrawal, please see our Privacy Policy.

Reviewer #1: Yes: Dr Lily Abedipour MD

Reviewer #2: No

Thank you

---

## [Decision Letter · Decision Letter 1]

10 Dec 2024

Dear Dr. Morshidi,

Thank you for submitting your manuscript to PLOS ONE. After careful consideration, we feel that it has merit but does not fully meet PLOS ONE’s publication criteria as it currently stands. Therefore, we invite you to submit a revised version of the manuscript that addresses the points raised during the review process.

**ACADEMIC EDITOR:** You have done a great job. However, it's better to incorporate some points like possible bias and clarification regarding the mediator and moderators. Therefore, please try to incorporate the comments raised by reviewers in your final submission.

We look forward to receiving your revised manuscript.

Kind regards,

Yadeta Alemayehu

Academic Editor

PLOS ONE

Journal Requirements:

Reviewers' comments:

Reviewer's Responses to Questions

**Comments to the Author**

Reviewer #3: (No Response)

Reviewer #4: (No Response)

Reviewer #5: All comments have been addressed

2. Is the manuscript technically sound, and do the data support the conclusions?

Reviewer #3: Yes

Reviewer #4: Yes

Reviewer #5: Partly

3. Has the statistical analysis been performed appropriately and rigorously?

Reviewer #3: Yes

Reviewer #4: Yes

Reviewer #5: Yes

4. Have the authors made all data underlying the findings in their manuscript fully available?

Reviewer #3: Yes

Reviewer #4: Yes

Reviewer #5: No

5. Is the manuscript presented in an intelligible fashion and written in standard English?

Reviewer #3: Yes

Reviewer #4: Yes

Reviewer #5: Yes

Reviewer #3: The results show that academic burnout had no significant mediator in most cases, the discussion would benefit from elaborating more on potential reasons for this unexpected outcome. This would help readers better understand the theoretical implications.

- The cross-cultural differences found, need further exploring, especially the unique patterns in Malaysia compared to Singapore and Australia. The discussion could be more richness if contextual factors (such as specific cultural pressures or educational systems) were included.

-A few sentences could benefit from grammatical corrections or restructuring for better readability.

Reviewer #4: (No Response)

Reviewer #5: regarding to improve the quality of this paper and for more details about this paper, please see the attached file for the comment.

**Do you want your identity to be public for this peer review?** For information about this choice, including consent withdrawal, please see our Privacy Policy

Reviewer #3: **Yes: ** Prof Dr Saad S Alatrany

Reviewer #4: **Yes: ** Aigerim Alpysbekova

Reviewer #5: No

---

## [Author Response · Author response to Decision Letter 2]

21 Apr 2025

RESPONSE TO REVIEWERS FOR PLOS ONE

We would like to thank the reviewer for the feedback and response towards the manuscript. Our response to the comments is as follows.

1. This statistic is outdated. Try to find updated data from around 2020 to ensure relevance and accuracy. For example, World Health Organization database had a suicide rate of 12.6 per 100,000 among 15 to 29-year-olds in the Western Pacific region in 2008.

The statistics have been updated with the line as follows:

World Health Organization database revealed a suicide rate of 17.18 per 100,000 among 15 to 29-year-olds in the Western Pacific region based on data collected in 2021 [5].

The citation has also been updated accordingly.

2. Although the introduction provides a global overview of youth suicide statistics, the transition to the core research question could be more specific. The current narrative should better highlight how this study fills a gap in understanding the specific mechanisms linking educational expectations to suicidal ideation. Refine the research question to clearly emphasize why investigating these mediators (academic burnout, psychache, and hopelessness) in this context is novel and valuable.

A revision that highlights how the study addresses the gap through the mediators has been added as follows:

Youth suicide is preventable and insight into the mechansims of suicide development is necessary to inform research and prevention. It is clear that high expectations can lead to negative psychological outcomes [19,25,26] which are manifestations of psychological pain and a sense of hopelessness [45,46]. We also identified academic burnout as both an outcome of high educational expectations (i.e., a demand beyond a student’s control) and a predictor of suicide ideation per the JDC [38]. As academic burnout is characterized by feelings of emotional exhaustion, cynicism, and inadequacy, we propose that academic burnout also predicts psychache and hopelessness which subsequently predicts suicidal ideation. Cynical beliefs about the purpose of studying along with the perceived incompetence as a student signify a pessimistic outlook and a sense of hopelessness toward their academic pursuit [16]. As such, the present study proposes a novel model that integrates the 3ST and the JDC in an attempt to examine the mechanism between educational expectations and suicide ideation through hypothesized a serial mediation model. Through this model, we empirically identified how burnout, psychological pain, and hopelessness predicts suicide risk with the goal of informing targeted suicide prevention in mitigating the burden of educational expectations among students in tertiary education.

3. The literature review, although informative, should engage more with existing research on academic stressors and suicide risk. Make sure there is a clear connection between past studies and the rationale for selecting academic burnout, psychache, and hopelessness as mediators. Integrate more theoretical backing or discuss prior findings on these mediators to justify their selection.

We believe that we have succinctly drawn connections between academic expectations and suicide risk by utilizing the Three-step theory of Suicide and the Job Demand Control model to fit within the journals word limit. The selection of mediators was informed by these models where i) psychache and hopelessness was drawn from the development of suicide ideation proposed in the 3ST and ii) burnout as a mediator between high demand and poor psychological outcomes as proposed by the Demand control model. We have noted this in The Present Study section of the manuscript.

4. The methodology is sufficiently detailed, but the use of convenience and snowball sampling may limit the generalizability of the findings. Acknowledge this as a limitation and suggest strategies for future studies to employ more representative sampling techniques. Clarify the ethical measures taken to ensure participant safety, especially considering the study's sensitive nature.

An acknowledgment of the sampling limitations was added in the Limitations section as follows:

Given the sampling limitations, caution should be taken in generalizing the findings. Future research should utilize a more representative technique to sample university student populations.

Additionally, an explanation about the ethical steps taken to ensure participant safety was added in the Methods section as follows:

Given the sensitive nature of the study, participants were made aware in the Information document about the risks of distress and were provided with contact details of free psychological support services in respective countries in the support document.

5. There is no mention of how missing data was handled, other than omitting incomplete responses, which could introduce bias. Include a brief discussion on whether imputation methods were considered for handling missing data and the potential impact of their omission on the results.

We have described in the Method section that incomplete or missing responses were omitted from the dataset prior to analysis. As such, no further imputation methods were considered as these missing cases were removed. We added a sentence in the Results to highlight this as follows:

Prior to analysis, the dataset was screened. Any incomplete responses were treated as missing completely at random and dropped from the final dataset. Note that possible bias may occur due to deletion by not accounting for reasons for missingness.

6. The scales used in this study show varied levels of internal consistency. Although the Psychache Scale has strong reliability (α = .92), the Higher Educational Expectation Scale (α = .70) is on the lower end of acceptability. Discuss the implications of employing instruments with different reliability levels and suggest comparing findings with alternative measures in future studies to reinforce conclusions.

To address this, we have included the reliability scores for each scale based on the data from this study across the three samples in the Method section. A summary for your reference is as follows:

Instrument Malaysia Singapore Australia

HEES .905 .912 .910

Psychache .958 .952 .957

BHS .841 .841 .828

SBI .886 .873 .886

SIS .922 .938 .945

Overall, the reliability scores were strong (>.80) for all.

7. The manuscript does not provide information on whether the scales were validated for use in multicultural contexts, such as Malaysia, Singapore, and Australia. Justify the choice of these instruments and note if any cross-cultural adaptations were made. Recommend future studies to conduct or include such validations to enhance cross-cultural reliability.

A description of the limitation regarding cross-cultural validation of the scale was added as follows:

Furthermore, cross-cultural validation and reliability of the instruments used in this study across Malaysia, Singapore, and Australia needs to be examined further.

8. The authors applied structural equation modeling (SEM) to examine their hypothesized model; however, the rationale for specifying 12 separate models is unclear. Provide an explanation of how each model contributes to the broader understanding of the research question. Include a discussion on the criteria and determination of model fit indices (e.g., RMSEA, CFI, TLI) to ensure thorough reporting.

We have revised the Proposed analysis section to provide more details on the specification of the model to explain how analysis were conducted.

The study employed a structural equation modelling with the maximum likelihood estimation using the Statistical Package for Social Science 27.0 AMOS 28. We specified the model with one exogenous variable (educational expectations), four endogenous variables (academic burnout, psychache, hopelessness, suicide ideation), and nine pathways based on the postulations by the 3ST and JDC theory resulting in an over-identified model (Fig 2). The same form (model) was tested with data for each source of educational expectation (self, parents, educator/institution, culture) respectively while data for the other variables remained constant. Testing the model for the different sources of expectations allows us to better identify how each source influences poor mental health outcomes. Multigroup analysis was performed in AMOS 28 on each source of expectation model with Group 1 (Malaysia), Group 2 (Singapore), and Group 3 (Australia). Estimates of model fit for each model include the comparative fit index (CFI), normed fit index (NFI), and goodness-of-fit statistic (GFI) values near or greater than 0.95, and a root mean square error of approximation (RMSEA) value near or less than 0.06 [52].

A pairwise parameter comparison using the critical ratio for differences on the nine specified pathways was examined. For a two-tailed test, a critical ratio difference z-score between the critical range of ±1.96 indicates no significant difference. Multigroup analysis allows us to determine cross-cultural variances on the effect of different types of expectations on poor mental health outcomes.

We have also restructured the Serial mediation analysis sub-section in the Results section to make it easier to describe the findings for each model and the model fit. Model fit indices were added for each model accordingly. We have reorganized the figures accordingly.

General response.

We sincerely appreciate the feedback and suggestions provided by Reviewer 3 regarding our manuscript. Suicide is a complex issue research on trying to comprehend what predicts an increased risk of suicide is highly debatable but vital. Multiple perspectives and theories exist in attempts to illuminate the underlying mechanisms in the development of suicidal outcomes. For our study, we specifically focused on addressing and providing insight into the link between high educational expectations and suicide ideation among students in higher education. Here, we utilized the 3ST and the JDC model to examine the mechanism linking expectations and suicide ideation development. We focused on academic expectation as a form of demand, leading to poor psychological outcomes such as academic burnout, psychache, and hopelessness (mediators), which in turn increases the risk of suicide ideation based on the postulations in these theoretical models. As such, we did not test other variables, such as personality, in this study. We have acknowledged that future research, however, should consider the role of other factors that may influence the pathways between expectations and suicide ideation. Given the theoretical postulations, we also did not examine burnout, psychache, and hopelessness as moderators, though we acknowledge that future investigation may examine how different levels of psychache and hopelessness may moderate the link between academic expectation, academic burnout, and suicidal ideation among students. The suggestions given are valuable input for suicide research, and we will continue to pursue our future investigations through a more comprehensive model.

The datasets and output supporting the conclusions of this article are available in figshare: https://figshare.com/projects/Education_expectation_and_suicide_ideation/217090

We would like to thank the reviewers for their time and input.

---

## [Decision Letter · Decision Letter 2]

22 Oct 2025

Dear Dr. Morshidi,

Thank you for submitting your manuscript to PLOS ONE. After careful consideration, we feel that it has merit but does not fully meet PLOS ONE’s publication criteria as it currently stands. Therefore, we invite you to submit a revised version of the manuscript that addresses the points raised during the review process.

We look forward to receiving your revised manuscript.

Kind regards,

Zheng Zhang

Academic Editor

PLOS ONE

Journal Requirements:

Reviewers' comments:

Reviewer's Responses to Questions

**Comments to the Author**

Reviewer #4: (No Response)

2. Is the manuscript technically sound, and do the data support the conclusions?

Reviewer #4: Partly

3. Has the statistical analysis been performed appropriately and rigorously?

Reviewer #4: Yes

4. Have the authors made all data underlying the findings in their manuscript fully available?

Reviewer #4: Yes

5. Is the manuscript presented in an intelligible fashion and written in standard English?

Reviewer #4: Yes

Reviewer #4: Thank you for your valuable contribution to the field. The topic is timely and relevant, and it is clear that significant effort went into data collection across multiple countries. This review focuses primarily on the Method section, which is foundational to the overall quality of the research. Several areas require clarification:

1. In the method section, the % of gender identification in the Malaysian sample do not add up to 100% (77.7% female + 21% male + 13% unidentified = 111.7%)

2. “Unidentified” is vague—does this refer to participants who chose not to disclose, non-binary participants, or missing data?

3. Explain the criteria for excluding responses due to missing data. Mention if any imputation methods were considered.

4. The section does not justify why the final sample sizes (224, 200, 217) are adequate for the analysis.

5. “At the time of data collection” is vague. The specific time period (e.g., semester, month/year) should be mentioned.

6. There is no mention of ethics approval from any Institutional Review Board (IRB) or Ethics Committee

7. What language(s) the survey was administered in? How long did it take participants to complete the survey?

**Do you want your identity to be public for this peer review?** For information about this choice, including consent withdrawal, please see our Privacy Policy

Reviewer #4: No

---

## [Author Response · Author response to Decision Letter 3]

26 Oct 2025

RESPONSE TO REVIEWERS

Review Comments to the Author and response;

Reviewer #4: Thank you for your valuable contribution to the field. The topic is timely and relevant, and it is clear that significant effort went into data collection across multiple countries. This review focuses primarily on the Method section, which is foundational to the overall quality of the research. Several areas require clarification:

Thank you for your time and feedback on the manuscript. We have responded to your comments in the following,

1. In the method section, the % of gender identification in the Malaysian sample do not add up to 100% (77.7% female + 21% male + 13% unidentified = 111.7%)

Thank you for highlighting this error. It appears we have missed the decimal point for the unidentified group, which was supposed to be 1.3% not 13%. This has been corrected.

See line 171, page 8 of Tracked Manuscript

2. “Unidentified” is vague—does this refer to participants who chose not to disclose, non-binary participants, or missing data?

A revised wording to better reflect the option of not disclosing (i.e., prefer not to say) has replaced the word ‘unidentified’ in the manuscript.

See lines 174, 176, and 178, page 8 of Tracked Manuscript

3. Explain the criteria for excluding responses due to missing data. Mention if any imputation methods were considered.

We have explained this process in the Preliminary Analysis section. Here, we noted that during data screening, incomplete responses were treated as missing and dropped from the final dataset. We also added that ‘… no imputation methods used…’.

See lines 250 – 251, page 12 of Tracked Manuscript.

4. The section does not justify why the final sample sizes (224, 200, 217) are adequate for the analysis.

An additional statement was provided as follows:

… The final sample size for each country was adequate for a moderate effect size in regression models (Wolf et al., 2013).

See lines 179 – 180, page 8 of Tracked Manuscript.

The reference list has also been updated to include Wolf et al. (2013).

5. “At the time of data collection” is vague. The specific time period (e.g., semester, month/year) should be mentioned.

A revised sentence that clearly mentioned the data collection period was added as follows:

… between July 2022 and December 2022.

See line 171, page 8 of Tracked Manuscript.

6. There is no mention of ethics approval from any Institutional Review Board (IRB) or Ethics Committee

A statement was added to the manuscript as follows:

… approved by the James Cook University Human Research Ethics Committee (H8552).

See line 225 – 2256, page 11 of Tracked Manuscript.

7. What language(s) the survey was administered in? How long did it take participants to complete the survey?

Additional details were added to the manuscript as follows.

The 15-minute study was in English…

See line 225, page 11 of Tracked Manuscript.

---

## [Decision Letter · Decision Letter 3]

14 Nov 2025

Uncovering the mechanism linking education expectation and suicide ideation among students in tertiary education: The mediating role of academic burnout, psychache, and hopelessness.

PONE-D-24-13596R3

Dear Dr.Izzat Morshidi,

We’re pleased to inform you that your manuscript has been judged scientifically suitable for publication and will be formally accepted for publication once it meets all outstanding technical requirements.

Kind regards,

Zheng Zhang

Academic Editor

PLOS ONE

Additional Editor Comments (optional):

Accept

Reviewers' comments:

Reviewer's Responses to Questions

**Comments to the Author**

Reviewer #6: All comments have been addressed

2. Is the manuscript technically sound, and do the data support the conclusions?

Reviewer #6: Yes

3. Has the statistical analysis been performed appropriately and rigorously?

Reviewer #6: Yes

4. Have the authors made all data underlying the findings in their manuscript fully available?

Reviewer #6: Yes

5. Is the manuscript presented in an intelligible fashion and written in standard English?

Reviewer #6: Yes

Reviewer #6: The manuscript demonstrates sound methodological rigor, relevance to the journal’s scope, and potential contribution to the understanding of psychological readiness in substance use treatment contexts.

Well done.

**Do you want your identity to be public for this peer review?** For information about this choice, including consent withdrawal, please see our Privacy Policy

Reviewer #6: **Yes: ** Gyanesh Kumar Tiwari

---

## [Editor Report · Acceptance letter]

PONE-D-24-13596R3

PLOS One

Dear Dr. Morshidi,

I'm pleased to inform you that your manuscript has been deemed suitable for publication in PLOS One. Congratulations! Your manuscript is now being handed over to our production team.

Kind regards,

on behalf of

Dr. Zheng Zhang

Academic Editor

PLOS One